# Reduction in and Preventive Effects for Oral-Cancer Risk with Antidepressant Treatment

**DOI:** 10.3390/jpm11070591

**Published:** 2021-06-23

**Authors:** Chia-Min Chung, Tzer-Min Kuo, Kun-Tu Yeh, Chien-Hung Lee, Ying-Chin Ko

**Affiliations:** 1Center for Drug Abuse and Addiction, China Medical University Hospital, Taichung 40447, Taiwan; Ibmsn141@gmail.com; 2Graduate Institute of Biomedical Sciences, China Medical University, Taichung 40447, Taiwan; 3Environment-Omics-Diseases Research Centre, China Medical University Hospital, China Medical University, No. 2 Yude Road, Taichung 40447, Taiwan; tzermin@hotmail.com; 4Department of Pathology, Changhua Christian Hospital, Changhua 50006, Taiwan; 10159@cch.org.tw; 5Department of Public Health, College of Health Science, Kaohsiung Medical University, Kaohsiung 80708, Taiwan; cnhung@kmu.edu.tw

**Keywords:** areca nut, antidepressants, oral submucous fibrosis, mouse model, betel quid, oral cancer

## Abstract

Areca nut (AN) was identified as carcinogenic to humans. Around 600 million people globally use AN in some form, yet no effective therapeutic drug is available to overcome AN addiction. This preclinical study examines the effects of antidepressants on AN use with animal models. We produced AN powder and dissolved it into drinking water, training 55 C57BL/6 mice in free self-selection to drink AN water or normal water. Then, the mice were randomly divided into four groups. Selective serotonin reuptake inhibitors (SSRIs), monoamine oxidase inhibitors (MAOIs), and *tricyclic antidepressants* (TCAs) were given as three treatment groups and one placebo group for four weeks. In the follow-up period, the preference and amount of free selection of AN and normal water, and oral pathological change were evaluated. There was a significant decrease in preference for AN drinking during the first four weeks, and the 36th week after drug withdrawal in the MAOI and SSRI groups (all *p* < 0.05). The drug-reducing effect of AN water in the 1–4-week period was significant in the MAOI group (*p* < 0.0001) and was also significant in the 3–4-week period in the SSRI group (*p* = 0.03). The TCA group did not show a decrease effect. At the endpoint (60 weeks), oral mucosal fibrosis (OSF) levels and risk in the SSRI (*p* = 0.0081) and MAOI (*p* = 0.01) groups were significantly lower than those in the control group. Antidepressant drugs MAOIs and SSRIs could reduce the amount of AN use and decrease the early stage of oral fibrosis in mice, but SSRIs may need to be boosted again.

## 1. Introduction

Betel-quid (BQ) chewing is common in South and Southeast Asia. It is estimated that, currently, 10% of the global population or nearly 600 million individuals regularly chew BQ [1,2]. Although BQ preparation methods vary among cultures and individuals, the areca nut (AN) is common component in all types of BQ preparation. The International Agency for Research on Cancer (IARC) has recognized the addictive properties of AN since 1985 and classified AN as carcinogenic to humans (Group 1) in 2004 [3]. AN consumption is associated with several diseases, such as oral potentially malignant disorders (OPMD) [4,5,6,7], oral-cavity and upper aerodigestive malignancies [8,9,10,11,12], metabolic syndromes [13], cardiovascular diseases [14], and adverse birth effects [15].

Winstock et al. reported that AN is the fourth most addictive substance [16]. Studies of its chemical constituents demonstrated that AN contains 0.15–0.67% alkaloids [17]. Psychoactive alkaloids, including the most abundant, arecoline and arecaidine, produce a high level of acetylcholine and monoamines (such as adrenaline and noradrenaline), and act as an agonist at muscarinic acetylcholine receptors [18,19]. This inhibits the reuptake of gamma-amino butyric acid (GABA) to induce anxiolytic effects. Previously reported research showed that regular BQ chewers have tolerance and a withdrawal syndrome [16,20]. Lee et al. conducted a large-scale survey across six Asian populations and found that BQ dependence prevalence among chewers largely varied across populations because of favorable BQ ingredients and culture-derived features. The one-year prevalence of dependence was 2.8–39.2% among the six Asian populations, and 20.9–99.6% of BQ chewers were BQ-dependent [6]. They also found that people who chewed BQ had high prevalence rates of OPMD, especially if they were dependent users [21,22,23].

Epidemiological data from several populations confirmed that AN is the major cause of oral submucous fibrosis (OSF) [24], and the proportions of OSF attributable to BQ use were 85% [25]. The study showed that 99 cases of OSF followed, and it found a 3.72% malignant-transformation rate for oral cancer (OC) with an average duration of 37.42 months [26]. BQ chewing was reported to be an independent risk factor, with an attributable risk accounting for 79% of patients with OC in Taiwan [25].

The cessation of AN consumption could be extremely difficult because of the tendency towards relapse and dependence. Quitting rates among AN users are relatively low. The quitting rate among the general population is 18%, and that for the indigenous population in Taiwan is 8% [27]. Several studies suggested that BQ chewers using tobacco or lime are less likely to quit BQ consumption [27,28].

Therefore, external assistance or even cessation therapy is necessary, yet no effective therapeutic drug is available to overcome BQ dependence.

Our previous study suggested that antidepressant use is associated with reduced risk of OC [29]. The potential mechanism is that antidepressant use may help to reduce or terminate BQ use, and consequently OSF risk. The limited understanding of the pharmacological basis of intoxication and AN use disorder means that there are no pharmacological replacement therapies for AN use disorder [30,31]. Therefore, in this study, we used animal models to examine the effect of antidepressants on decreasing BQ consumption and reducing OC risk.

## 2. Materials and Methods

### 2.1. Reagents and Areca-Nut Extract (ANE) Preparation

Fresh ANs were purchased from local commercial stores in Tainan, Taiwan. ANs were weighed, crushed, and incubated at 37 °C for 2 h. Fibers were removed with two layers of gauze and 90 m filters. The aqueous ANE was collected and condensed to powder form, and then stored at −20 °C. Dosing solutions were prepared by dissolving the appropriate amount of ANE powder into distilled water. According to a previous study, lime-containing BQ users have a significantly higher dependence domain of unsuccessful cut-down [21]. We dissolved 1 g of lime in 1000 mL of distilled water (1 mg/mL). The AN dosages were based on the health effects of years of chewing and daily consumption reported in previous studies [5,25]. Groups of daily consumption over 20 AN and 20 years of the chewing habit had significant risk for OSF or oral leukoplakia. Assuming an average human body weight of 60 kg and an average daily consumption of 20 AN, the dose was 0.25 g/mL. Considering that the mean body weight of mice during the study period remained at around 30 g the average dose of AN was approximately 2 mg per day. The average daily consumption of water for mice is approximately 4 mL. The concentration of the AN solution was 500 ug/mL with 1 mg/mL lime.

### 2.2. Drug Preparation

Treatment with antidepressants was administrated via AN water, which mimicked the condition of human AN consumption, and could reduce the effects of the withdrawal symptoms of BQ chewing. Antidepressant drugs were given at the following doses: imipramine (10 mg/kg /day); tricyclic antidepressants (TCA), citalopram (15 mg/kg/day); selective serotonin reuptake inhibitor (SSRI) and moclobemide (25 mg/kg/day); monoamine oxidase inhibitors (MAOI). All drugs were purchased from Sigma-Aldrich (St Louis, MO, USA) and totally administrated in 4 consecutive weeks. The doses of the antidepressant solution were calculated as mg/kg for a mouse dose. Imipramine, citalopram, and moclobemide were separately dissolved in distilled water and AN water, then diluted to the desired concentration in one bottle. The doses of antidepressant agents and the time interval between AN cessation administration were modified from the results of various research groups [32,33,34].

### 2.3. Experimental Procedure

Six-week-old C57BL/6 male mice (*n* = 55) were used, purchased from the National Laboratory Animal Center (NLAC, Taiwan), with a rearing-environment temperature of 20–25 °C and relative humidity of 60–67%. After adaptive feed for 4 weeks, the mice were randomly divided into 4 groups: the MAOI, SSRI, and TCA treatment groups, and the placebo group (AN water). There were ten mice in the SSRI group, 15 mice in the MAOI group, 5 mice in the TCA group, and 25 mice in the placebo group. Placebo groups without applying antidepressants treatment as control groups were observed to assess effects of the consumption amount of areca-nut water and OSF levels in parallel with the use of MAOI and SSRI groups. Free-selection drinking was first used to feed AN water and normal drinking water for 4 weeks. Second, the preference ratio of AN water to normal drinking water was evaluated. Two bottles of water that contained normal drinking water and AN water, respectively, in the same cage were freely selected by mice. The consumption amount was measured for 4 consecutive weeks as the baseline data. A ratio of the amount of AN water consumed to that of normal drinking water greater than 1 was defined as a higher AN preference. Third, treatment with antidepressants was administered via AN water. Fourth, preference-level changes after antidepressant treatment were compared. After the 36-week follow-up, we evaluated the relapse change of AN consumption amount after treatment. According to the literature, the prevalence and risk of OSF increased with chewing duration and daily consumption [5]. Mice were consecutively fed AN water until 60 weeks. After the 60-week follow-up, mice were sacrificed, and pathological changes in oral fibrosis rate and levels were observed. More details are shown in Figure 1. In this study, animal procedures conformed to the guidelines published by the National Institute of Health (NIH publication no. 85-23) and were approved by the Institutional Animal Care and Use Committee (IACUC) of the China Medical University.

### 2.4. Immunohistochemistry Analysis

Tongues from the treated mice were fixed in 4% paraformaldehyde for 3 days and then embedded in paraffin. Then, 3 or 1.5 μm sections were stained with hematoxylin and eosin (H&E) for histological analysis. For the assessment of fibrosis potential, tongue sections were stained with Masson’s trichrome by using a Trichrome Modified Masson’s Stain Kit (Scytek Labs, Logan, UT, USA). Tongue fibrosis sections from all mice in each group were counted and scored under 100× magnification. The quantitative measurement (fibrosis index %) was performed using ImageJ software (version 1.50b) software. OSF cases and severity of precancerous changes were determined by the above averaged fibrosis index (≥5). The histopathological classification of OSF patients is suggested in several studies on the basis of fibroblastic response, hyalinized collagen, blood vessel morphology, and lymphocyte infiltration [35,36]. In this study, the results of all the immunohistochemical tests were interpreted by two pathologists.

### 2.5. Statistical Analysis

Statistical analysis was performed with SAS 9.2 software (SAS Institute, Cary, NC, USA). Data are presented as mean ± standard deviation. Multiple sets of average comparison were examined using one-way ANOVA. Changes within and between groups were analyzed using paired-sample *t* tests and independent-sample *t* tests, respectively. Categorical and binary variables were compared using the χ^2^ test or Fisher exact test. Mean changes in AN consumption regarding the amount and preference ratio were compared before and after the antidepressant treatment within groups, then analyzed between groups by MANOVA. Interaction *p* values are presented for antidepressant treatment * time (before, and after 2, 4, and 36 weeks) effect. Generalized estimating equation (GEE) analyses were performed to examine mean changes from the baseline in repeated-measure assessments with treatment, follow-up, and treatment–follow-up interaction. We analyzed the associations of OSF risk with different antidepressant-treatment groups using the Kaplan–Meier method and the Cox proportional-hazards regression model.

## 3. Results

### 3.1. Safety of Antidepressants in Animal Study

We conducted an animal study to examine the antidepressant effects on the reduction in AN consumption. We chose those three antidepressant drugs because of their known pharmacokinetics and safety profiles. The effect of long-term AN treatment on mouse body weight was measured every week. We further compared body-weight changes among the MAOI, SSRI and placebo groups. Among these groups, body weight increased with time, but it was not significantly different (*p* = 0.7985; Appendix A). Long-term follow-up for mouse death rate was not significant between the treatment and placebo groups (Fisher exact *p* = 0.8372).

### 3.2. Change in Consumption Ratio and Amount of Areca-Nut Water after Antidepressant Treatment

We then evaluated the changes in the preference ratio of AN water after long -term antidepressant treatment. Preference ratio was defined as the consumption amount of AN water divided by the amount of normal drinking water in a free-selection condition. Formula = consumption amount of AN water (average mL per week)/consumption amount of normal drinking water (average ml per week). Results are presented in Table 1 (top), showing a decreased consumption ratio (12.44%) in the Placebo A group (*p* = 0.16). After the 36-week follow-up, the consumption ratio was decreased by 17.49% in the MAOI treatment (*p* = 0.04) and 42.81% in the SSRI treatment group (*p* = 0.02) compared to before treatment. The consumption ratio decreased by 17.43% in the Placebo A group (*p* = 0.12). The TCA and Placebo B groups were not significantly decreased after antidepressant treatment (Table 1, bottom). Because the TCA group was not significantly decreased in preference ratio compared to the placebo group, the follow-up period was halted at four weeks after treatment. More detailed changes are presented in Appendix A.

We performed MANOVA to evaluate changes in consumption amount (average mL/week) of AN water after antidepressant treatment in the four- and 36-week follow-ups. Average consumption amounts per week at baseline in the placebo, MAOI, and SSRI group were 21.7 ± 4.92, 19.6 ± 4.19, and 18 ± 11.6, respectively. There was no significant difference among the groups at the baseline (*p* = 0.385). The antidepressant treatment × time interaction represents the main between-group comparison in the consumption-amount changes across time (Table 2). The results show that there was a statistically significant difference between the MAOI treatment and placebo groups at the two- and four-week follow-ups (*p* = 0.001 and *p* = 0.0175). After the 36-week follow-up, there was no significant difference between the MAOI treatment and placebo group. However, there was statistically significant difference between the SSRI treatment and placebo groups at the four- and 36-week follow-ups (*p* = 0.0091 and *p* = 0.0454), but there was no significant difference between the MAOI treatment and placebo group at the two-week follow-up. Repeated-measures ANOVA for antidepressant treatment showed significant differences regarding time, and the interaction between follow-up time and MAOI treatment group (*p* = 0.0002 at two weeks, *p* < 0.0001 at four weeks; *p* = 0.0001 at 36 weeks). The interaction between the follow-up time and SSRI treatment was significantly decreased at the four-week follow-up (*p* = 0.003). A GEE model was established to examine the interaction between antidepressant treatment and follow-up time. Results of GEE analysis were similar to those in the interaction between time and drug effects with MANOVA (Appendix A).

### 3.3. Difference in Fibrosis Levels and Risk between Treatment and Placebo Groups

H&E-stained tongue specimens from the antidepressant-treatment and placebo groups showed normal tongue features and mouse OSF. Figure 2A shows a photomicrograph showing OSF (blue) in mouse tongue tissue. Fibrosis indices are shown in the bar graphs between the study groups. The MAOI and SSRI groups had significantly lower fibrosis index values (Figure 2B; *p* = 0.01 and *p* = 0.0081, respectively). Moreover, the fibrogenesis process was protected in the SSRI treatment group compared to in the placebo group (HR = 0.08, 95% CI: 0.01–0.76; and HR = 0.19, 95% CI: 0.05–0.7; respectively, *p* < 0.01; Table 3). Figure 3 shows the Kaplan–Meier curve for fibrosis risk rate according to treatment group. After the 60-week follow-up, the fibrosis risk rate for the MAOI and SSRI groups was significantly lower than that for the placebo group (*p* = 0.0407 and 0.0016, respectively).

## 4. Discussion

MAOI and SSRI treatment decreased AN consumption and OSF risks. The antidepressant properties of BQ and arecoline were reported to directly inhibit MAO-A enzymatic activity in cell, rat, and human models [37,38,39,40]. MAO-A metabolizes various primary, secondary, and tertiary monoamines, and preferentially deaminates neurotransmitters relative to depression. MAOI is used in the treatment of clinical depression and may be used for BQ cessation. Spring et al. found that fluoxetine, an SSRI, initially increased smoking cessation among smokers with a history of depressive disorders [41]. SSRIs were shown to inhibit serotonin reuptake by blocking serotonin transporters, thus probably decreasing the risk of smoking initiation and nicotine dependence, and increasing the likelihood of success of smoking cessation [42]. Arecoline is an important AN ingredient and has a comparable chemical structure to that of nicotine. SSRI treatment reduced AN consumption in our animal model. The potential mechanism is related to the effect of dopamine crosstalk with the release of serotonin in the brain, and there is strong evidence that serotonergic tone plays a role in the effects of addiction. Some studies reported that therapy with nortriptyline, a TCA, appears to be effective in the treatment of nicotine addiction [43,44]. However, in this study, TCA treatment did not significantly reduce the consumption amount of AN compared to in the placebo group.

OSF is a chronic disorder characterized by fibrosis of the mucosa lining the upper digestive tract involving the oral cavity, oro- and hypopharynx, and upper third of the esophagus, predominantly seen in people of South and Southeast Asia [45]. The main etiologic agent causing OSF was confirmed to be compounds and fibers in AN. The IARC classified ‘BQ without tobacco’, ‘BQ with tobacco’, and AN as carcinogenic to humans (Group 1) [3]. Cigarette smoking significantly contributes to the risk of leukoplakia, but does not contribute to OSF. BQ-chewing users with OSF experienced a higher risk at each exposure level of chewing duration, quantity, and cumulative measure than that of those who had leukoplakia [4,5,25]. The malignant transformation rate of OSF was reported to be over 2% per year [46]. Several studies were recently conducted to determine the possible mechanisms involved in malignant transformation. Until now, no single molecular pathway has been identified that is either necessary or sufficient for the development of fibrosis. This bars any molecularly targeted therapies. Because AN plays a major etiologic role in OSF, the cessation of AN use remains pivotal in the management of this disorder. Our study demonstrated that MAOI and SSRI treatment could reduce oral fibrosis levels via decreasing the consumption amount of AN in the mouse model.

MAOI treatment significantly decreased the consumption amount of AN at the two- and four-week follow-up, but no significant decrease was observed at 36 weeks. The interaction between follow-up time and MAOI treatment was significantly decreased at the two-, four-, and 36-week follow-up. However, SSRI treatment significantly decreased the amount of AN consumed at the four- and 36-week follow-ups. The interaction between the follow-up time and SSRI treatment was only significantly decreased at the four-week follow-up. The study of Hung et al. demonstrated that a reduction in daily BQ consumption after antidepressant therapy, including MAO-A inhibitors and SSRIs, was observed among patients with depression [47,48]. Our study and previous clinical trials provide preliminary evidence for appropriate treatments for the cessation of AN addiction.

SSRI treatment also has more protective effects on reducing OSF risk compared to those from the MAOI treatment. Serotonin is a stimulator of tissue fibrosis [49] and binds the 5-HT_2B_ receptor on fibroblasts, leading to fibroblast activation. Serotonin and the 5-HT_2B_ receptor are associated with fibrosis [50]. The agonism of 5-HT_2B_ was implicated in fibrosis caused by fenfluramine, used in the treatment of obesity [51] and psychiatric disorders [52], both of which trigger 5-HT_2B_ signaling. Dopamine agonists with structural similarity to 5-HT, such as pergolide and cabergoline, which are used in the treatment of Parkinson’s disease, are also associated with the development of fibrosis in heart valves involving 5-HT_2B_ agonism [53]. In addition to SSRIs, Alsamman et al. demonstrated that tricyclic antidepressants reduce hepatic fibrosis by inhibiting acid ceramidase [54].

To our knowledge, this is the first animal study to report that SSRIs can reduce the risk of oral fibrosis. The association and potential mechanisms between SSRIs and inhibition of oral fibrosis need to be investigated. SSRI treatment has long-term effects on decreasing the consumption amount of AN compared to those of MAOI treatment because SSRIs have a synergic effect on reducing the consumption amount of AN and inhibiting fibrosis activation. SSRIs may need to be boosted again, or the dose may need to be increased to efficiently reduce AN consumption.

This study has limitations. First, no serum and urine markers for metabolites of AN were utilized as evidence of cessation. The effect of cessation was neither validated nor measured in the mouse model. We only presented the decreased consumption preference ratio and amounts of AN. We cannot present the complete cessation rate in this study. Second, the consumption amounts of AN were not significant in the TCA treatment, which may have been due to the small sample size in the TCA group. Third, clinical indices for the assessment of liver fibrosis are well-established. However, clinical histopathological criteria for other fibrosis diseases have rarely been developed. In this study, a mouse model with long-term areca-nut treatment for the very early stage of oral fibrosis was employed. Accordingly, we used software to measure collagen deposition and assess the effects of antidepressants on oral fibrosis. We demonstrated that antidepressants could reduce oral fibrosis levels and risk in experimental subgroups in cross-sectional data. Without investigating proper control groups of experimental animals and limited sample sizes, we cannot examine the changes of fibrosis levels in the different doses of antidepressants and time duration.

## 5. Conclusions

After long-term follow-up, MAOI and SSRI treatments could reduce AN consumption and decrease very early stage oral fibrosis in a mouse model. These results may allow clinicians to provide appropriate treatment for the cessation of AN consumption and provide alternative ways to reduce OSF and oral-cancer risk.

## Figures and Tables

**Figure 1 jpm-11-00591-f001:**
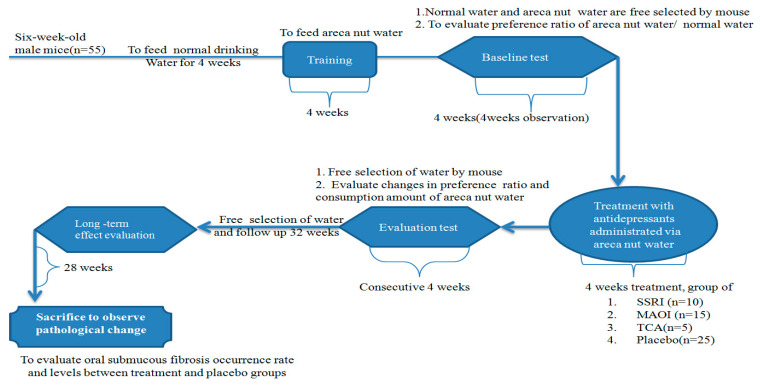
Schematic of evaluation of areca-nut use with antidepressant treatment in animal model.

**Figure 2 jpm-11-00591-f002:**
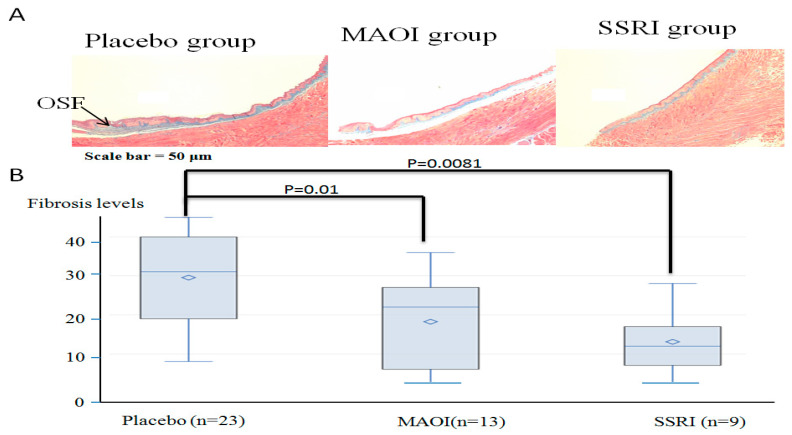
(**A**) H&E (upper) trichrome of oral submucous fibrosis in mouse tongue tissue. Photomicrograph showing oral submucous fibrosis (blue) in mouse tongue tissue. Scale bar = 50 μm. Quantification of fibrosis levels obtained from ImageJ software. (**B**) Fibrosis levels treated by MAOI and SSRI compared to placebo group.

**Figure 3 jpm-11-00591-f003:**
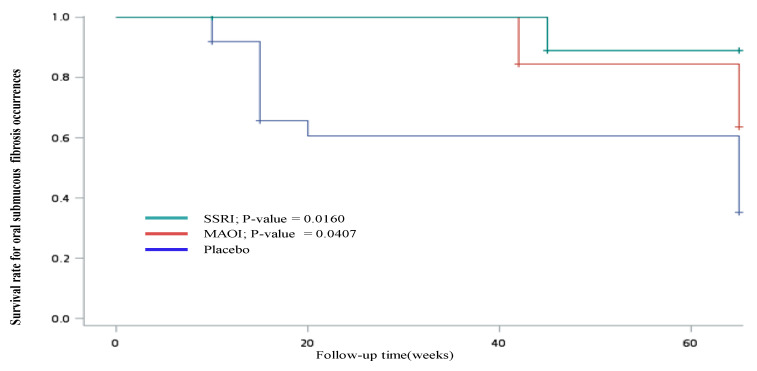
Effects of areca-nut use on oral submucous fibrosis risk after antidepressant treatment analyzed with Kaplan–Meier method in animal model.

**Table 1 jpm-11-00591-t001:** Changes in weekly preference ratio (±SD) with consumption of area nut water after antidepressants treatment in mouse mode (follow-up at 4 and 36 weeks).

					Changes in		Changes in	
Groups	N	Baseline	4-Week Follow-Up	36-Week Follow-Up	4-Week Follow-Up (%)	*p* Value	36-Week Follow-Up (%)	*p* Value
Placebo A	15	1.63 (0.38)	1.43 (0.47)	1.34 (0.49)	12.2	0.16	17.8	0.12
MOAI	15	1.43 (0.39)	0.70 (0.55)	1.18 (0.15)	51.0	<0.01	17.5	0.04
SSRI	10	1.58 (0.94)	1.07 (0.34)	0.90 (0.33)	32.3	0.03	42.8	0.02
Placebo B	10	1.34 (0.36)	1.22 (0.32)	-	8.96	0.26	-	-
TCA	5	1.19 (0.44)	1.08 (0.30)	-	9.24	0.24	-	-

Preference ratio defined that consumption amount of areca nut water was divided by the amount of normal drinking water in free-selection condition. Formula = consumption amount of areca nut water (mL/weekly)/consumption amount of normal drinking water (mL/weekly). TCA group was not significantly decreased in preference ratio compared to placebo group. Follow-up period was halted at 4 weeks after treatment. SD: standard deviation.

**Table 2 jpm-11-00591-t002:** Changes in consumption amount (averaged mL/per week) of area nut water after antidepressants treatment in mouse mode (follow-up 4 and 36 weeks).

				MAOI				SSRI			
Scale	Placebo, Mean ± SD (*n*)	MOAI ± SD (*n*)	SSRI, Mean ± SD (*n*)	F	*p*	F ^a^	*p* ^a^	F	*p*	F ^a^	*p* ^a^
Baseline	21.7 ± 4.92 (15)	19.6 ± 4.19 (15)	18 ± 11.6 (10)	0.34	0.5656	0.39	0.537	0.46	0.5023	1.02	0.3265
Weeks 1–2	26.2 ± 4.19 (15)	6.7 ± 8.96 (15)	26.05 ± 11.4 (10)	16.94	0.0003	14.3	0.001	0.1	0.9763	0.79	0.3879
Weeks 3–4	21.0 ± 6.30 (15)	7.4 ± 5.08 (15)	8.2 ± 4.3 (10)	11.14	0.0024	6.6	0.0175	8.06	0.0093	8.66	0.0091
Weeks 36	25.1 ± 5.84 (15)	19.2 ± 4.15 (15)	15.7 ± 5.46 (10)	1.96	0.1749	0.5	0.4867	4.81	0.0418	4.66	0.0454
Drug											
Week 1–2 × drug				16.83	0.0003	18.28	0.0002	0.69	0.4137	0.3	0.8714
Week 3–4 × drug				12.53	<0.0001	13.33	<0.0001	3.46	0.0398	3.62	0.0351
Week 36 × drug				12.2	<0.0001	8.16	0.0001	1.53	0.2175	0.98	0.4118

MANOVA was performed to examine interaction between antidepressant intervention × time effects (before and after treatment—36 week follow up). ^a^ Mean changes were analyzed between two groups (MAOI vs placebo; SSRI vs placebo) by MANOVA with adjustment for mice weight.

**Table 3 jpm-11-00591-t003:** Hazard ratio for oral submucous fibrosis occurrence in different treatment groups compared with placebo group.

		Fibrosis Index (≧5)		
	N (Expired)	No	Yes	HR (95%CI)
Number of mice	45 (5)			
Groups				
Placebo	23 (2)	9 (39.1)	14 (60.9)	1
MAOI	13 (2)	9 (69.2)	4 (30.8)	0.29 (0.06–1.21)
SSRI	9 (1)	8 (88.9)	1 (11.1)	0.08 (0.01–0.76)
MAOI + SSRI	22 (3)	17 (77.3)	5 (22.7)	0.19 (0.05–0.70)

## Data Availability

The data presented in this study are available on request from the corresponding author.

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
