# Peer review of "Reduction in and Preventive Effects for Oral-Cancer Risk with Antidepressant Treatment"

_jpm, 2021, doi:10.3390/jpm11070591_

Round 1

Reviewer 1 Report

Chung et al. investigated the effects of antidepressants on areca nut (AN) consumption in a preclinical murine model. Investigating the same animals, they also determined semiquantitatively the degree of oral submucous fibrosis (OSF) in the tongue by using immunochemistry. Using this experimental design, they aimed at testing three different hypotheses: 1) (Some) antidepressants decrease the preference for and thereby consumption of AN-fortified water to plain water in mice. 2) (Some) antidepressants decrease the level of OSF in the murine model. Finally) (Some) antidepressants may decrease the risk of developing OSF by decreasing AN consumption.

Authors were successful in confirming the first two hypotheses.

1) They successfully demonstrated that MAOI and SSRI significantly decreased the preference for AN-fortified water consumption. This result seems to be new in the murine model. However, beneficial effects of antidepressants have already been demonstrated in the clinical (i.e. human) setting. See e.g. Hung CC et al. Medicine 2020; 99:1. and Ko AMS et al. Progress in Neuropsychopharmacol & Biol Psychiatry 2020; 103:109982. Moreover, in the Abstract they wrote (lines 30-32): “The drug reduce effect of amount 30 of AN water was also significantly through 36 weeks in MAOI group (p<0.0001), but only during the 3-4week was observed in SSRI group (p=0.035).” (In addition, this sentence suffers from several grammatical and typographic errors!) In contrast, in Discussion they wrote (lines 261-262): “ … results reveal that SSRI treatment has long term effects on decreasing consumption amount of AN 261 compared to MAOI treatment.” This obvious contradiction between the two statements should be clarified.

2) Indeed, this study is the first to report that MAOI and SSRI decreased the level of AN-induced OSM in mice.

3) However, without investigating proper control groups of experimental animals, one cannot accept that “MAOI and SSRI treatment” … reduced “the risks of OSF risk” (Sic! Why to use plural here and why to repeat “risk” in this sentence? – a grammatical-stylistic question of the reviewer) “via decreasing consumption amount” (Sic! What did authors mean by “consumption amount”?) “of AN in mice model.” The observed reduction of the degree of fibrosis can be related to a) a direct effect of antidepressants – as antifibrotic effects of antidepressants has already been demonstrated independently of betel quid or AN consumption in mice (e.g. Xin X et al. Biomed Pharmacother 2019; 115:108870.; Alsamman S et al. Sci Transl Med 2020; 12:easy8798.), or b) as authors suggested, because of the reduction in AN consumption of mice treated with MAOI and SSRI. However, to prove this latter possibility, authors should have demonstrated that decreasing (as controlled by the experimental setting w/o applying antidepressants) AN-fortified water consumption to the degree observed in parallel with the use of MAOI and SSRI, did or did not have a similar effect on the level of OSM as in the experimental subgroups.

In Materials and Methods authors wrote (lines136-137) that: “Severity of pre-cancerous changes was determined …” but they did not define what did they understand under the term “precancerous changes” nor how they defined their “severity”.

English language, style, grammar and spelling requires a thorough check-up and correction throughout the text.

Author Response

Chung et al. investigated the effects of antidepressants on areca nut (AN) consumption in a preclinical murine model. Investigating the same animals, they also determined semiquantitatively the degree of oral submucous fibrosis (OSF) in the tongue by using immunochemistry. Using this experimental design, they aimed at testing three different hypotheses: 1) (Some) antidepressants decrease the preference for and thereby consumption of AN-fortified water to plain water in mice. 2) (Some) antidepressants decrease the level of OSF in the murine model. Finally) (Some) antidepressants may decrease the risk of developing OSF by decreasing AN consumption.

Authors were successful in confirming the first two hypotheses.

  1. They successfully demonstrated that MAOI and SSRI significantly decreased the preference for AN-fortified water consumption. This result seems to be new in the murine model. However, beneficial effects of antidepressants have already been demonstrated in the clinical (i.e. human) setting. See e.g. Hung CC et al. Medicine 2020; 99:1. and Ko AMS et al. Progress in Neuropsychopharmacol & Biol Psychiatry 2020; 103:109982. Moreover, in the Abstract they wrote (lines 30-32): “The drug reduces the effect of amount 30 of AN water was also significantly through 36 weeks in MAOI group (p<0.0001), but only during the 3-4week was observed in SSRI group (p=0.035).” (In addition, this sentence suffers from several grammatical and typographic errors!)

Authors’ Response:

Thank you for your comment. We have revised the comments of the reviewer. We send the manuscript to English editing to improve grammar and correct spelling mistakes. Please see the attached file for the certificate of English editing

  1. In contrast, in the Discussion they wrote (lines 261-262): “ … results reveal that SSRI treatment has long term effects on decreasing consumption amount of AN 261 compared to MAOI treatment.” This obvious contradiction between the two statements should be clarified.

Authors’ Response:

Thank you for your comment. We emphasized long-term treatment effects of SSRI are better than MOAI treatment. We have rewritten the sentence.

  1. However, without investigating proper control groups of experimental animals, one cannot accept that “MAOI and SSRI treatment” … reduced “the risks of OSF risk” (Sic! Why to use plural here and why to repeat “risk” in this sentence? – a grammatical-stylistic question of the reviewer) “via decreasing consumption amount” (Sic! What did authors mean by “consumption amount”?) “of AN in mice model.

Authors’ Response:

Thank you. We have rewritten the sentence.

  1. Indeed, this study is the first to report that MAOI and SSRI decreased the level of AN-induced OSM in mice. The observed reduction of the degree of fibrosis can be related to a) a direct effect of antidepressants – as antifibrotic effects of antidepressants has already been demonstrated independently of betel quid or AN consumption in mice (e.g. Xin X et al. Biomed Pharmacother 2019; 115:108870.; Alsamman S et al. Sci Transl Med 2020; 12:easy8798.), or b) as authors suggested, because of the reduction in AN consumption of mice treated with MAOI and SSRI. However, to prove this latter possibility, authors should have demonstrated that decreasing (as controlled by the experimental setting w/o applying antidepressants) AN-fortified water consumption to the degree observed in parallel with the use of MAOI and SSRI, did or did not have a similar effect on the level of OSM as in the experimental subgroups.

Authors’ Response:

We thank the reviewer for the valuable suggestions. Clinical indexes for the assessment of liver fibrosis have been well established. However, the clinical histopathological criteria for other fibrosis diseases were rarely developed so far. We demonstrate that antidepressants can reduce in AN-fortified water consumption and has synergic effect on reducing consumption amount of AN and inhibiting fibrosis activation. We just present cross-section data to examine differences in fibrosis levels and risk in experimental subgroups (Placebo, MAOI, and SSRI groups). Please see figure2 and figure3

  1. English language, style, grammar and spelling requires a thorough check-up and correction throughout the text.

Authors’ Response:

Thank you for your comment. We have revised manuscripts. We send the manuscript to English editing to improve grammar and correct spelling mistakes. Please see the attached file for the certificate of English editing

Reviewer 2 Report

This is a preclinical study that examines the effects of anti-depressants on areca nut use using animal models. It is a very interesting topic and a well-designed study.

However, the following suggestions should be addressed:

In the abstract section, the meaning of abbreviations SSRI, MAOI, and TCA have to be specified.  

In material and methods section does not clarify the different histopathological criteria (grades) for OSF. It would be interesting to have this information and add these findings to the results section. Furthermore, it is necessary to explain how the fibrosis index was obtained (lines 134,135). With the text information, it not possible to understand this index.

In the discussion section, more limitations should be pointed out such as the small sample size in TCA group, and others. Also, due to saliva is the fluid of the oral cavity authors should include it in the paragraph of limitations.

In the conclusion section, no references should be included. You have to indicate only the conclusions of the present study. Then, the sentence “The study of Lee…” has to be removed. The sentence “our results showed…” (lines 282,283) has the same meaning as the sentence “our study reveals…” (lines 280,281). The conclusion should be rewritten.

At the end of this paper, the authors should check if a figure or table is missing.

Author Response

This is a preclinical study that examines the effects of anti-depressants on areca nut use using animal models. It is a very interesting topic and a well-designed study.

However, the following suggestions should be addressed:

  1. In the abstract section, the meaning of abbreviations SSRI, MAOI, and TCA have to be specified.

Authors’ Response:

Thank you for your comment. We have defined the meaning of abbreviations in the abstract section. . Please see Line34-35, page2.

  1. In material and methods section does not clarify the different histopathological criteria (grades) for OSF. It would be interesting to have this information and add these findings to the results section. Furthermore, it is necessary to explain how the fibrosis index was obtained (lines 134,135). With the text information, it not possible to understand this index.

 Ans: We thank the reviewer for the valuable suggestions.

Clinical indexes for the assessment of liver fibrosis have been well established. However, the clinical histopathological criteria for other fibrosis diseases were rarely developed so far. Indeed, histopathological classification of OSF patients is suggested in several studies based on the fibroblastic response, hyalinized collagen, blood vessel morphology, and lymphocyte infiltration. Please see the reference:

  • Shruti Pandya, Ajay Kumar Chaudhary, Mamta Singh, Mangal Singh, and Ravi Mehrotra. Correlation of histopathological diagnosis with habits and clinical findings in oral submucous fibrosis. Head Neck Oncol. 2009, 1:10
  • Sudharani Basawaraj Biradar, Anita Dnyanoba Munde, Basawaraj Chanabasappa Biradar, Safia Shoeb Shaik, Shweta Mishra. Oral submucous fibrosis: A clinico-histopathological correlational study. J Cancer Res Ther. 2018,14(3):597.

Collagen deposition is the very early stage of oral fibrosis.

In this study, a mouse model with long-term areca nut treatment for the very early stage of oral fibrosis was employed. We, therefore, used the software to measure collagen deposition to assess the effects of antidepressants on oral fibrosis. We have explained the fibrosis index in the method section. Please see Line191-195, page 9.

  1. In the discussion section, more limitations should be pointed out such as the small sample size in TCA group, and others. Also, due to saliva is the fluid of the oral cavity authors should include it in the paragraph of limitations.

Authors’ Response:

Thank you for your comment. We have included more limitations in the limitation section. Please see Line342-348, page 17.

  1. In the conclusion section, no references should be included. You have to indicate only the conclusions of the present study. Then, the sentence “The study of Lee…” has to be removed. The sentence “our results showed…” (lines 282,283) has the same meaning as the sentence “our study reveals…” (lines 280,281). The conclusion should be rewritten.

Authors’ Response:

Thank you. We have rewritten the conclusion. Please see Line 350-353, page 18.

  1. At the end of this paper, the authors should check if a figure or table is missing.

Authors’ Response:

Thank you for your comment. We have submitted some figures and tables in the supplementary materials. Please see supplementary figures and tables

Reviewer 3 Report

It is very intresting topic in Oral cancer, so more infromations needed to support the ain of the study, also there is no immuohoistochemical figures shown, neither controls, I was wondering how did you do immunohistochemistery?

Author Response

  1. It is very intresting topic in Oral cancer, so more infromations needed to support the ain of the study, also there is no immuohoistochemical figures shown, neither controls, I was wondering how did you do immunohistochemistery?

Authors’ Response:

 Thank you for your valuable suggestions.

  1. The hematoxylin and eosin (H&E) stain of the tongue from all experimental mice for histology analysis was performed. We have added the selective and paired results in supplementary figure 4. Please see supplementary figure4.
  2. Indeed, a result of a negative correlation between the expressions of fibrosis markers including COL1A1 or fibronectin and antidepressant effects will be helpful in strengthening the evidence for the study. Staining protocol optimization for immunohistochemical staining using those antibodies is ongoing.

Round 2

Reviewer 1 Report

The revision, as provided by the authors cannot be properly  evaluated.

First, because authors did not reply to this reviewer's comments in a point-by-point fashion. Eg. it is not enough to reply "Thank you for your comment. We emphasized long-term treatment effects of SSRI are better than MOAI treatment. We have rewritten the sentence." and "We have rewritten the sentence." but they should write down the changes implemented in the text in their response as well. Second, a revised manuscript in its final form has not been provided, only the hard-to-read version full of corrections indicated with strike through characters and the use of two different colors in addition to black characters. Such way it is not possible to properly evaluate the revised version. Even this way it can be judged that the manuscript still contains grammatical errors. Eg. in the sentence in lines 101-105 where the subjective "alkaloids" is in the plural form but the predicates "acts (as AN agonist)", and "inhibits" is written in the singular form. I think it is not the job nor the competence of the reviewer to find all grammatical and stylistic errors and to correct them. Most importantly, authors did not address the major criticism raised by this reviewer, i.e. without investigating proper control groups of experimental animals, one cannot accept that “MAOI and SSRI treatment” … reduced “the risks of OSF risk” (Sic! Why to use plural here and why to repeat “risk” in this sentence? – a grammatical-stylistic question of the reviewer) via decreasing consumption amount”. Authors corrected the obvious linguistic errors but they did not respond to the professional criticism and in the revised manuscript they still concluded that (lines 530-531) "MAOI and SSRI treatment could reduce the risks of OSF risks via decreasing the consumption amount of AN in the mice mouse model" although this statement, as pointed out in the original review, was not investigated by implementing proper controls.

Author Response

The revision, as provided by the authors cannot be properly evaluated.

  1. First, because authors did not reply to this reviewer's comments in a point-by-point fashion. Eg. it is not enough to reply "Thank you for your comment. We emphasized long-term treatment effects of SSRI are better than MOAI treatment. We have rewritten the sentence." and "We have rewritten the sentence." but they should write down the changes implemented in the text in their response as well.

Authors’ Response:

We do apologize that we did not respond reviewer's comments clear. Thank you very much for allowing us to revise our manuscript again. We have made all necessary changes according to the reviewers’ comments. We have marked the change with red color in the revised manuscript. Please see the attached file for the revised manuscript.

  1. Second, a revised manuscript in its final form has not been provided, only the hard-to-read version full of corrections indicated with strike through characters and the use of two different colors in addition to black characters. Such way it is not possible to properly evaluate the revised version. Even this way it can be judged that the manuscript still contains grammatical errors. Eg. in the sentence in lines 101-105 where the subjective "alkaloids" is in the plural form but the predicates "acts (as AN agonist)", and "inhibits" is written in the singular form. I think it is not the job nor the competence of the reviewer to find all grammatical and stylistic errors and to correct them.

Authors’ Response:

Thank you for your comment. We have changed corrections and provided a clear revised version. We have corrected these grammatical errors. We have marked the changes with red color. Please see lines 102-106, page4.

  1. Most importantly, authors did not address the major criticism raised by this reviewer, i.e. without investigating proper control groups of experimental animals, one cannot accept that “MAOI and SSRI treatment” … reduced “the risks of OSF risk” (Sic! Why to use plural here and why to repeat “risk” in this sentence? – a grammatical-stylistic question of the reviewer) via decreasing consumption amount”. Authors corrected the obvious linguistic errors but they did not respond to the professional criticism and in the revised manuscript they still concluded that (lines 530-531) "MAOI and SSRI treatment could reduce the risks of OSF risks via decreasing the consumption amount of AN in the mice mouse model" although this statement, as pointed out in the original review, was not investigated by implementing proper controls.

Authors’ Response:

We have addressed the limitation of proper control groups of experimental animals in the study. Please see the statement as follows.”We demonstrated that antidepressants can reduce fibrosis levels and risk in experimental subgroups in cross-sectional data. Without investigating proper control groups of experimental animals and limited sample sizes, we cannot examine the changes of fibrosis levels in the different doses of antidepressants and time duration”. We have marked the changes with red color. Please see line368-376, page17-18.

We have toned down the conclusion as follows: “The effects of antidepressants decreased very early stage of oral fibrosis”. We have marked the changes with red color. Please see line378-379, page18.

Previously reviewer’ comment:

  1. In contrast, in Discussion they wrote “ … results reveal that SSRI treatment has long term effects on decreasing consumption amount of AN 261 compared to MAOI treatment.” This obvious contradiction between the two statements should be clarified.

Authors’ Response:

Indeed, we found this contradiction. These controversies may come from the dose of antidepressants, treatment duration, and inhibition of fibrosis, and so on.

We have clarified two statements as follows: “SSRI treatment has long-term effects on decreasing the consumption amount of AN compared to those of MAOI treatment because SSRIs have a synergic effect on reducing the consumption amount of AN and inhibiting fibrosis activation. SSRIs may need to be boosted again, or the dose may need to be increased to efficiently reduce AN consumption.” We have marked the changes with red color.  Please see lines 352-356, page17.

  1. However, without investigating proper control groups of experimental animals, one cannot accept that “MAOI and SSRI treatment” … reduced “the risks of OSF risk” (Sic! Why to use plural here and why to repeat “risk” in this sentence? – a grammatical-stylistic question of the reviewer) “via decreasing consumption amount” (Sic! What did authors mean by “consumption amount”?) “of AN in mice model.

Authors’ Response:

We are sorry that we did not make a clear response regarding the proper control group. We have defined control groups clear in the Methods section as follows.“Placebo groups without applying antidepressant treatment as control groups were observed effects on consumption amount of areca-nut water and OSF levels in parallel with the use of MAOI and SSRI groups”. We have marked the changes with red color. Please see lines 172-174, page8.

We did have not proper control groups of experimental animals in the study to control potential confounders, such as treatment duration of antidepressants, different doses of antidepressants, and so on. We have addressed limitations as follows: We demonstrated that antidepressants could reduce oral fibrosis levels and risk in experimental subgroups in cross-sectional data. Without investigating proper control groups of experimental animals and limited sample sizes, we cannot examine the changes of fibrosis levels in the different doses of antidepressants and time duration. We have marked the changes with red color. Please see lines368-376, page17-18.

We have rewritten the sentence as follows: “Our study demonstrated that MAOI and SSRI treatment could reduce oral fibrosis levels via decreasing the consumption amount of AN in the mouse model.” We have marked the changes with red color. Please see line315-317, page15.
